# Hybrid Systems of Gels and Nanoparticles for Cancer Therapy: Advances in Multifunctional Therapeutic Platforms

**DOI:** 10.3390/gels11030170

**Published:** 2025-02-26

**Authors:** Kibeom Kim

**Affiliations:** Department of Chemistry and Life Science, Sahmyook University, Seoul 01795, Republic of Korea; kibumsy@syu.ac.kr

**Keywords:** gel, nanoparticle, cancer therapy

## Abstract

Cancer is a global health concern. Various therapeutic approaches, including chemotherapy, photodynamic therapy, and immunotherapy, have been developed for cancer treatment. Silica nanoparticles, quantum dots, and metal–organic framework (MOF)-based nanomedicines have gained interest in cancer therapy because of their selective accumulation in tumors via the enhanced permeability and retention (EPR) effect. However, bare nanoparticles face challenges including poor biocompatibility, low stability, limited drug-loading capacity, and rapid clearance by the reticuloendothelial system (RES). Gels with unique three-dimensional network structures formed through various interactions such as covalent and hydrogen bonds are emerging as promising materials for addressing these challenges. Gel hybridization enhances biocompatibility, facilitates controlled drug release, and confers cancer-targeting abilities to nanoparticles. This review discusses gel–nanoparticle hybrid systems for cancer treatment developed in the past five years and analyzes the roles of gels in these systems.

## 1. Introduction

Cancer remains a major problem and a worldwide global health issue [1,2,3,4,5]. Various therapeutic approaches have been developed to combat cancer, including chemotherapy (CHT) [6,7,8,9], photodynamic therapy (PDT) [10,11,12], photothermal therapy (PTT) [13,14,15], immunotherapy (IMT) [16,17,18], and radiotherapy (RT) [19,20,21]. CHT is widely used in cancer treatment due to its effectiveness even with relatively low drug dosages. However, most CHT drugs have hydrophobic properties, making it difficult to deliver sufficient quantities of these drugs to cancer cells [22,23,24,25]. Additionally, the nonspecific accumulation of CHT drugs causes side effects toxic to normal cells [26,27,28,29,30]. Drug delivery systems have been developed [31,32,33], and recently, nanoparticles such as silica nanoparticles (SNP) [34,35,36], gold nanoparticles (AuNPs) [37,38,39], quantum dots (QDs) [40,41,42], polymeric nanoparticles [43,44], and metal–organic frameworks (MOFs) have gained increasing attention as drug delivery vehicles [45,46,47,48]. These nanoparticle-based drug delivery systems enable specific accumulation in cancer cells through the enhanced permeability and retention (EPR) effect, enhancing the efficacy of cancer therapy [49,50,51,52]. However, depending on the characteristics of the nanoparticles, drug delivery systems still face challenges such as low capacity for drug loading, inadequate biocompatibility, clearance by the reticuloendothelial system (RES), difficulties in controlled drug release, and insufficient specificity for cancer cell targeting [26,53,54,55,56,57,58,59,60,61].

Gels can address the challenges associated with NP-based drug delivery systems [62,63]. Gels possess three-dimensional network structures formed by covalent bonds, ionic interactions, hydrogen bonds, and van der Waals forces [64,65,66]. This type of network structure can encapsulate liquids, endowing gels with the unique properties of both solids and liquids [67,68]. Gels maintain a soft yet stable structure and can be engineered into various shapes and functions by controlling their physical and chemical properties [69,70]. These properties of gels enhance their biocompatibility, facilitate controlled drug release, and improve cancer targeting by drug delivery systems [71,72,73].

This review summarizes research over the past five years on cancer treatment systems using hybrid structures integrating gels and nanoparticles. This innovative hybridization method overcomes the challenges of nanoparticle-based drug delivery systems, such as low biocompatibility, low drug capacity, and limited tumor-targeting specificity [73,74,75]. The role of the gel in these hybrid systems is categorized into five areas: enhanced tumor retention and localized drug administration, improvement of stability and biocompatibility of nanoparticles, controlled drug release, active cancer targeting, and the role of nanoparticle type in hybrid gel–nanoparticle systems.

## 2. Gels and Nanoparticle Hybridization for Cancer Therapy

### 2.1. Enhanced Tumor Retention and Localized Drug Administration

Nanoparticles enhance the efficacy of cancer therapy by targeting cancer tissues via the EPR effect, depending on their size. However, a significant portion of nanomedicine is cleared by the RES, reducing its therapeutic effects [76,77]. Therefore, strategies to extend the retention time of nanoparticles in tumor tissues are essential for enhancing their therapeutic efficacy. Gel possesses high adhesive properties through interactions with tumor tissue, enhancing the retention time in the tumor via hybridization with nanoparticles. By injecting the gel–nanoparticle hybrid directly into the tumor site or inducing it to form a hydrogel within the tumor, the retention time of the nanoparticles can be extended, facilitating targeted drug delivery [78,79,80,81,82]. This section highlights studies that extend intratumoral retention time and facilitate localized drug delivery through hydrogel–nanoparticle hybrid systems.

Wang et al. developed an innovative system based on a condensation reaction between bovine serum albumin (BSA) nanoparticles, O-phthalaldehyde (OPA), and N-nucleophile to improve tissue adhesion and achieve controlled drug release (Figure 1a) [83]. This system immobilizes the drug at the tumor site, promoting drug accumulation in the tumor tissue and inhibiting drug leakage to normal tissues, therapy increasing the therapeutic efficacy and reducing side effects (Figure 1b).

In that study, paclitaxel (PTX) was loaded onto BSA-based nanoparticles. The BSA nanoparticles were synthesized using the desolvation method, through which PTX was encapsulated via physical interactions with albumin. The synthesized nanoparticle size was 120~160 nm and they had a ζ-potential of approximately −15.2 to 10.9 mV. After synthesis, the PTX-loaded nanoparticles (PTX@BN) were mixed with branched polyethylene glycol (branched PEG) and OPA functional groups and underwent a condensation reaction with the primary amine of BSA to form a PTX@BN-loaded hydrogel (PTX@BN gel) in situ. After synthesis, the PTX-loaded nanoparticles (PTX@BN) were mixed with branched polyethylene glycol PEG containing OPA functional groups. PEG is a synthetic, hydrophilic polymer with high biocompatibility, and it can enhance the biocompatibility of gel when incorporated into its structure. This mixture was then subjected to a condensation reaction between OPA and the primary amine of BSA to form a PTX@BN-loaded hydrogel (PTX@BN@Gel). The PTX@BN@Gel adhered to tumor tissue, maintained local drug concentration, and facilitated sustained drug release.

The mechanical properties, biodegradability, and biocompatibility of PTX@BN@Gel were evaluated. In addition, C26 colon cancer cells and 4T1 breast cancer cells were incubated with PTX@BN@Gel to assess their anticancer effects. PTX@BN@Gel was injected into C26 and 4T1 tumor-bearing mouse models for in vivo experiments. In the results, PTX@BN@Gel demonstrated improved inhibition of tumor growth and increased survival rates compared with free PTX. These results suggest that PTX@BN@Gel has excellent therapeutic efficiacy (Figure 1c–e).

Veloso et al. developed a lipogel system that controlled the release of two different types of anticancer drugs using distinct nanoparticles [84]. This system consisted of Mn-based magnetic nanoparticles (MNPs), doxorubicin (DOX)-loaded liposomes, and PTX-loaded mesoporous silica-coated gold nanorods (MS-AuNRs). The hydrodynamic diameter of the liposome was 100 nm, and its ζ-potential was −2 mV. Lipogels were synthesized through the buffer switch method using 2-Naph-L-Phe-Z-ΔPhe-OH (compound 1 or C1) and 2-Naph-L-Phe-Z-ΔPhe-FRGDG-OH (compound 2 or C2), with C2 containing a cancer-targeting peptide sequence (Phe-Arg-Gly-Asp-Gly, FRGDG). Liposomes and MS-AuNRs were encapsulated within the lipogels, which controlled the release of DOX under a high-frequency alternating magnetic field (HF-AMF) and the release of PTX under near-infrared (NIR) irradiation. The gel in this system served as a drug carrier, acting as a scaffold for the system, promoting sustained drug release, and enhancing drug accumulation in the tumor tissue. The lipogel’s morphology exhibited a fiber-like structure, with fiber thickness ranging from 20 to 30 nm depending on the ratio of C1 to C2. The system was incubated with HEK293 (human embryonic kidney cells) and SHSY5Y (human neuroblastoma) spheroid model cells to observe its therapeutic effects. The drug-free lipogel treated group demonstrated no toxicity, while the cancer drug-loaded lipogels exhibited high toxicity under HA-AMF and NIR irradiation. In particular, SHSY5Y cells were more sensitive to toxicity, while the HEK293 cells were relatively more resistant and showed comparatively lower toxicity. In contrast to other systems that release drugs in response to a single stimulus, this multimodal system released different drugs via HA-AMF and NIR stimulation. This dual stimuli-based system demonstrated a higher therapeutic effect through sustained drug release and reduced drug resistance compared with single-modal treatments. The drug release mechanism involved the release of DOX by increasing the mobility of the liposome bilayer through localized heat generated by MNPs under HF-AMF, while PTX was released by altering the physical properties of the gel network via the heat generated by the MS-AuNRs under NIR irradiation. That study demonstrated the potential of a multimodal stimuli-responsive system to enhance the effectiveness of cancer therapy.

Ito et al. developed a system using superparamagnetic iron oxide nanoparticles (SPIONs) and decyl-group-modified Alaskan pollock gelatin (C10-ApGltn)-based microparticles (C10MPs). Gelatin is a protein-based natural polymer obtained through the hydrolysis of collagen. It has the advantages of high biocompatibility and excellent biodegradability. Additionally, it undergoes a reversible gel–sol transition in water. C10MPs form a colloidal gel in a wet environment and encapsulate SPIONs to enhance tissue adhesion and therapeutic efficacy [85]. C10-ApGltn was synthesized by reacting the amine group of Alaskan pollock gelatin (ApGltn) with the aldehyde group of decanal. The synthesized C10-ApGltn formed a coacervate in the solvent, which was transformed by thermal crosslinking into C10MPs of 2.3 ± 1.0 μm. When a mixture of SPIONs and C10MPs was sprayed onto cancer tissue, the C10MPs formed a colloidal gel through hydrophobic interactions under the moist conditions and stably encapsulated the SPIONs through coordination bonds. The encapsulated SPIONs facilitated hyperthermal therapy by generating localized heat under an AMF, effectively removing cancer cells. SPION/C10MP and SPION/OrgMP Alaskan pollock gelatin-based microparticles lacking decyl groups were compared in order to evaluate tissue adhesion. SPIONs/OrgMPs have low adhesion strength owing to the lack of hydrophobic interactions, whereas SPIONs/C10MPs exhibit an adhesion strength approximately three times higher than that of SPIONs/OrgMPs. To evaluate therapeutic efficiency, human mesenchymal stem cells (hMSCs) and human colon cancer cells (KM12-Luc) were treated with SPIONs/C10MPs. Cytotoxicity was low prior to exposure to AMF but increased after exposure. Cancer cells exhibited higher toxicity than normal cells. SPIONs/C10MPs were administered to KM12-Luc-bearing mice for in vivo experiments. The SPIONs/C10MPs-treated group demonstrated inhibited tumor growth and prolonged survival under AMF conditions. These results suggest that SPIONs/C10MPs may be effective in the treatment of cancer.

Zeng et al. developed a hydrogel system composed of MOFs and bisphosphonate-functionalized (BP) HA-BP for cancer treatment [86]. To overcome the challenges of conventional CHT, including difficulties in tumor targeting and severe side effects, a local drug delivery system was developed for direct injection into the tumor site. DOX was loaded into the MOFs during synthesis, and the MOFs were degraded at a low pH and high concentrations of adenosine triphosphate (ATP). This degradation promoted drug release. The ζ-potential of the synthesized MOF increased from −10.14 mV to 0.016 mV after DOX encapsulation. DOX-loaded MOF (MOF@DOX) and HA-BP formed the HA-BP·MOF@DOX hydrogel via dynamic coordination bonds. HA-BP·MOF@DOX can facilitate drug accumulation in tumor tissue and minimize systemic exposure and side effects. DOX was effectively released from the MOF at low pH and a high concentration of ATP. Comparing drug release in a neutral environment (pH 7.4) without ATP and in an acidic environment (pH 5) with 2 mM ATP, the acidic environment resulted in a 21-fold increase in drug release. HA-BP·MOF@DOX was administered to CT-26 colon cancer cells to evaluate its therapeutic effect, which was analyzed using a CCK-8 assay and caspase staining. HA-BP·MOF@DOX demonstrated decreased cell viability and increased cell apoptosis. In addition, HA-BP·MOF@DOX and MOF@DOX were administered to tumor-bearing mice to compare the therapeutic effects of the gel. The HA-BP·MOF@DOX-administered group demonstrated enhanced inhibition of tumor growth and an increased survival rate. Furthermore, the non-toxicity of HA-BP·MOF@DOX was evaluated through histopathological analysis, and weight loss and side effects were found to be minimized. The research demonstrated that MOF-based hydrogels can be utilized as localized drug delivery systems for cancer therapy.

Jiang et al. developed a hydrogel consisting of palladium nanosheets (Pd NSs) and thiol-terminated four-arm polyethylene glycol (4arm-PEG-thiol) without additional crosslinking agents (Figure 2a) [87]. Pd NSs possess a high photothermal conversion efficiency of 52% under 808 nm NIR light and can be easily surface-modified through palladium–sulfur (Pd-S) bonding. This property of the Pd NSs facilitated the formation of a hydrogel (Pd Gel) with 4arm-PEG-thiol. Furthermore, a DOX-loaded hydrogel (Dox@Pd Gel) was synthesized by adding DOX during the hydrogel formation (Figure 2b). The Dox@Pd Gel possessed shear-thinning properties allowing its injection into the body using a syringe. After injection, the hydrogel remained stable in the tumor tissue, increasing drug accumulation at the tumor site. Additionally, Pd NSs in the hydrogel generated heat, increasing the localized temperature to 38 °C and facilitating controlled drug release. The Pd gel and Dox@Pd gels were applied to murine breast cancer cells (4T1 cells) for in vitro experimentation under NIR irradiation. The Pd gel-treated group exhibited minimal toxicity before NIR irradiation, whereas after NIR irradiation, cell viability decreased by up to 57%. The Dox@Pd gel-treated group exhibited 90% cell viability before NIR irradiation, whereas after NIR irradiation, cell viability decreased by up to 20%. This result implies a synergistic effect of chemo-photothermal combination therapy. The Dox@Pd gel was administered to a tumor-bearing mouse model to evaluate inhibition of tumor growth. Administration of the Dox@Pd gel resulted in increased survival rates and significantly reduced tumor size without a significant decrease in weight. This result indicated the effective cancer therapeutic potential of the Dox@Pd gel as well as its high biocompatibility. That study demonstrated the potential of a system consisting of Pd NS and 4arm-PEG-thiol for cancer therapy.

### 2.2. Improvement of Stability and Biocompatibility of Nanoparticles

Nanoparticles have large surface areas that facilitate the loading of various drugs. In addition, their unique properties enable various therapies such as PTT and PDT [88,89,90]. However, bare nanoparticles have limitations, including plasma protein adsorption (protein corona), colloidal instability, and immune system-triggered inflammatory responses [91,92,93]. To mitigate these problems, nanoparticles can be functionalized with various surface ligands [94,95,96]. This section discusses gel and nanoparticle hybridization to enhance stability and biocompatibility.

Shen et al. developed a tumor-immunosuppressive-microenvironment-modulating hydrogel (TIMmH_sd_) consisting of semiconducting polymer nanoparticles (SPIIN) with a ζ-potential of −9.04 mV, an immunoadjuvant cytosine–phosphorothioate–guanine oligodeoxynucleotide (CpG ODNs), and a Ca^2+^-responsive alginate hydrogel (ALG) for combined PTT and IMT (Figure 3a) [97]. Alginate is a natural polysaccharide that exhibits excellent biocompatibility and biodegradability and is widely used in medicine, drug delivery, and tissue engineering. The system maximized the synergistic effect of combining PTT and IMT (Figure 3b). A 1064 nm, 1.0 W/cm^2^ system was developed to overcome the traditional challenges of IMT, including low cellular uptake, rapid degradation, and insufficient accumulation of CpG ODNs at the tumor site. In the body, the carboxylic acid groups of ALG and Ca^2+^ ions form a stable hydrogel through coordination bonding. This hydrogel formation in the body facilitates localized drug accumulation and sustained release of CpG ODNs. Under NIR irradiation (1064 nm, 1.0 W/cm^2^), TIMmH_sd_ increases the localized temperature up to 55 °C, facilitating tumor treatment through PTT. Furthermore, the morphology of TIMmH_sd_ is maintained even after repeated NIR irradiation. TIMmH_sd_ was administered to 4T1 breast cancer cells for in vitro experiments. Under NIR irradiation, the TIMmH_sd_-administered group demonstrated decrease in cell viability of up to 16%. This result indicated a potent PTT effect. Additionally, PTT induces immunogenic cell death (ICD) and increases the expression of calreticulin (CRT) and high-mobility group box 1 (HMGB1), enhancing immune stimulation. TIMmH_sd_ was administered to tumor-bearing mice at different sites to observe the IMT effect. The TIMmH_sd_-treated mice showed not only a decrease in the size of the treated tumors but also a reduction in the size of non-treated tumors. This result implies the potential to induce a systemic immune response (Figure 3c,d). In addition, no toxicity was observed in major organs such as the liver, kidneys, and heart, and only minimal weight loss was noted, proving the high biocompatibility of TIMmH_sd_. That study demonstrated the potential of a hydrogel-based therapeutic method to facilitate the combination of CHT and IMT through sustained drug release and activation of immune response.

Sadeghi et al. developed a novel therapeutic hydrogel system (ADHG) consisting of dopamine-grafted alginate (Alg-DA). Polydopamine nanoparticles (PDA NPs) were grown within the hydrogel (Figure 4) [98]. The DA NP-encapsulated ADHG (ADHG-PDAN) had high tissue-adhesion, biocompatibility, and photothermal conversion properties, promoting the development of effective cancer therapeutics through PTT. Alg-DA, constituting the ADHG, was synthesized via an EDC/NHS coupling reaction between the carboxylic acid group (-COOH) of alginate and the amine group (-NH_2_) of dopamine, leading to the formation of an amide bond. The synthesized Alg-DA formed the ADHG through electrostatic interactions with Ca^2+^ ions via crosslinking after being placed in a mold, frozen, and immersed in a CaCl_2_ solution. The synthesized ADHG was immersed in a dopamine solution to generate PDA NPs inside the gel, induced through self-oxidation of dopamine. When irradiated with NIR (808 nm, 1.0 W/cm^2^), the temperature of the ADHG-PDAN hydrogel increased to 42.4 °C. To confirm the effect of PTT, ADHG-PDAN was administered to MDA-MB-231 breast cancer cells. The ADHG-PDAN-administered cells demonstrated a decrease in cell viability of up to 42.2% under NIR irradiation. After PTT, DOX was added to the ADHG-PDAN-treated cells, which showed lower cell viability (32.6%) than the DOX- only group (51.6%). This was attributed to the PTT’s effect on increasing the cell membrane permeability of DOX. The research demonstrated the biocompatibility and tissue adhesion of ADHG-PDAN, showing its potential to enhance the effectiveness of cancer treatments. 

Zhang et al. developed a hydrogel-based cancer therapy system consisting of zeolitic imidazolate framework-8 (ZIF-8) and gelatin methacryloyl (GelMA) to prevent recurrence and promote wound healing after tumor resection [78]. ZIF-8 is stable at neutral pH (7.4) but degrades under the acidic conditions of the tumor microenvironment (pH 5.5–6.0), inducing drug release. DOX was loaded into ZIF-8 during its synthesis, and the surface of the ZIF-8 was modified with cerium oxide (CeO_2_) nanoparticles, acting as reactive oxygen species (ROS) scavengers, to form ZC-DOX. The synthesized ZC-DOX had a particle size of 102 nm and a ζ-potential of 24.2 mV. GelMA was synthesized through the reaction between the lysine residues of gelatin and methacrylic anhydride. The synthesized GelMA was able to form a gel in the presence of a photoinitiator and under UV irradiation. ZC-DOX was encapsulated in a hydrogel to generate ZC-DOX@GEL. ZC-DOX demonstrated fast drug release at tumor pH, whereas hydrogel-encapsulated ZC-DOX@GEL exhibited sustained release due to the increased stability provided by the hydrogel. ZC@GEL and ZC-DOX@GEL were administered to 4T1 breast cancer cells, and their therapeutic effects were compared. The ZC@GEL-treated group exhibited 90% cell viability, while the ZC-DOX@GEL-treated group exhibited a decrease in cell viability of up to 40%. In addition, 4T1 cell migration was inhibited by ZC-DOX@GEL, as confirmed by a cell migration inhibition assay. To confirm the antioxidant and anti-inflammatory effects, ZC@GEL was administered to mouse bone marrow-derived macrophages (BMDMs) with ROS production induced using H_2_O_2_ and LPS, and the intracellular ROS levels were effectively reduced. To compare the therapeutic effects in vivo, DOX@GEL and ZC-DOX@GEL were administered to tumor-bearing mice. The ZC-DOX@GEL-treated group demonstrated a high rate of inhibition of tumor growth and excellent wound-healing effects. Additionally, ZC-DOX@GEL increased collagen production and promoted tissue regeneration, as confirmed by H&E and Masson’s trichrome staining. These results indicate that effective treatment using the ZC-DOX@GEL system is possible through the removal of tumor cells and the alleviation of inflammation.

Ziaei et al. developed a hydrogel consisting of chitosan (CS)-modified AuNPs (CS/AuNPs), oxidized alginate (OAL), and gelatin [99]. Chitosan is a natural polysaccharide derived from the chitin of crustaceans, and has received much attention for use in drug delivery systems due to its high biocompatibility and enzymatic degradability in the body. The surface of AuNPs was modified with CS, which then formed a CS gel through ionic crosslinking between the amine groups (-NH_2_) of CS and the anionic groups (PO_4_^3−^) of sodium triphosphate pentabasic (TPP). CS not only facilitated the loading of DOX but also increased the stability and biocompatibility of the AuNPs. DOX-loaded CS/AuNPs (CS/AuNPs-DOX) were added to a solution consisting of OAL, gelatin, and beta-glycerophosphate (β-GP). This solution was transformed into the CS/AuNPs-DOX-encapsulated hydrogel (OALGH) through a Schiff base reaction between the aldehyde (-CHO) groups of OAL and the amine (-NH_2_) groups of the gelatin. β-GP is an ionic crosslinker that controls the hydrogel pH from acidic to neutral and induces gelation by accelerating the sol-to-gel transition. The synthesized OALGH had a particle size of 209 nm and a ζ-potential of 19.2 mV. The OALGH hydrogel had a high capacity for water absorption and slow degradation properties. The release was controlled by the pH-dependent solubility. DOX solubility increased in the tumor environment (pH 5.8), which enhanced drug release, whereas under normal physiological conditions (pH 7.4), drug release was slow. Thus, the pH-dependent solubility of DOX facilitated tumor-selective drug delivery. OALGH was administered to MCF-7 breast cancer cells to determine its therapeutic effects. OALGH-treated cells showed decreased cell viability. In addition, AuNPs as radiation sensitizers not only facilitate radiotherapy but also enable tumor monitoring through their optical properties. The AuNPs and DOX inside the hydrogel enhanced the therapeutic effect through synergistic effects. That study demonstrated the potential of OALGH as a cancer therapy.

### 2.3. Controlled Drug Release

Gels are highly biocompatible and can control drug release in response to external stimuli. In particular, temperature changes induce the gel-to-sol transition; thus, the hybridization of gels with nanoparticles that induce photothermal effects can be effectively utilized in drug delivery systems [71,100,101,102,103]. For example, the hybridization of iron nanoparticles or AuNPs with gels facilitates controlled drug release through magnetic field changes or NIR irradiation [104,105]. Furthermore, gels can transform their network structure in response to the acidic tumor environment, which induces changes in the porosity of the gel and promotes drug release [106,107]. By leveraging these properties, gel–nanoparticle hybridization controls drug release and enhances the effectiveness of cancer therapy. This section introduces studies on controlled drug release through gel–nanoparticle hybridization.

Jia et al. developed an NIR-responsive hydrogel consisting of a PDLLA-PEG-PDLLA (PLEL) hydrogel and multifunctional nanoparticles (RIC NPs) to prevent recurrence and metastasis after breast cancer surgery (Figure 5a) [108]. These RIC NPs possess a photothermal effect that facilitates PTT, and the PLEL responds to temperature, which facilitates controlled drug release (Figure 5b). RIC NPs were synthesized by mixing indocyanine green (ICG), resiquimod (R848), and CpG ODNs solution via self-assembly. Each component served an important role in PTT and immune activation. However, the three functional components in RIC NPs have low solubility in water and decompose rapidly; therefore, hybridization with a hydrogel was used to address this challenge. PLEL is a triblock copolymer that was synthesized through the ring-opening polymerization of D,L-lactide. In this process, stannous octoate was used as a catalyst, and polyethylene glycol was used as an initiator. The synthesized PLEL was physically mixed with RIC NPs, which maintain a liquid state at room temperature while transforming into a gel at physiological temperature (37 °C). PLEL self-assembles into micelles, which further aggregate to form a 3D micelle network, ultimately leading to hydrogel formation. In addition, the system temperature increases to 55 °C under NIR irradiation, leading to a secondary gel-to-sol transition. This secondary gel-to-sol transition facilitates localized drug release control and induces the release of R848 and CpG ODNs, which trigger a potent antitumor immune response (Figure 5c,d). PLEL hydrogel was administered to 4T1 and RAW 264.7 cells to assess biocompatibility. The PLEL-administered group exhibited low toxicity, with 80% cell viability. Meanwhile, PLEL hydrogel was administered to tumor-bearing mice, demonstrating a strong therapeutic effect under light irradiation, in the study of an NIR-responsive drug release system consisting of gel and nanoparticle hybridization.

Wang et al. developed a stimuli-responsive injectable antibacterial hydrogel (CP@Au@DC_AC50) to treat uveal melanoma (UM) [109]. CP@Au@DC_AC50 consisted of a CS–puerarin (CP) hydrogel incorporating gold nanorods (AuNRs) and DC_AC50. Puerarin forms a hydrogel through self-assembly via a rapid heating-cooling process; however, the puerarin hydrogel had a rigid nanofiber structure, posing a challenge for direct injection. Therefore, a simple one-step grinding method was used to synthesize a self-assembled antibacterial CP hydrogel by combining CS and puerarin, thereby enabling the formation of an injectable puerarin hydrogel. CS, which has excellent biocompatibility, biodegradability, and antibacterial properties, along with puerarin, AuNRs, and the gene-targeting therapeutic drug DC_AC50, was mixed with acetic acid and ground until the acid was completely volatilized. After grinding, the CP@Au@DC_AC50 hydrogel was synthesized through various interactions, including hydrogen bonding, π–π interactions, hydrophobic interactions, van der Waals forces, and electrostatic interactions. The AuNRs enhanced the mechanical strength of the hydrogel and facilitated the controlled release of DC_AC50 by inducing PTT and a temperature-responsive gel–sol transition under NIR irradiation. The CP@Au@DC_AC50 hydrogel had a self-assembled nanofiber structure and exhibited shear-thinning and self-healing properties, making it suitable for intraocular injection therapy. DC_AC50 is a gene therapy agent that targets the ATOX1 protein, and its release is promoted by the photothermal effects of AuNPs. The CP@Au@DC_AC50 hydrogel showed a gel–sol transition at temperatures above 53 °C, and upon NIR irradiation, cancer cell death through drug release and PTT was observed. CP@Au@DC_AC50 was administered to OCM1 and OM431 UM cells to observe its effects on cancer. The CP@Au@DC_AC50 group showed a decrease in cell viability of up to 21% under NIR irradiation. In addition, when CP@Au@DC_AC50 was administered to a tumor-bearing mouse model, postoperative recurrence suppression and antibacterial effects were observed. That study presented an NIR-responsive gel–nanoparticle hybrid system with controlled drug release.

Zhang et al. developed a NIR- and pH-responsive hydrogel system consisting of polydopamine (PDA)-coated Fe_3_O_4_ nanocubes (Fe_3_O_4_@PDA) and a CS-based hydrogel [110]. Fe_3_O_4_@PDA nanocubes possess high photothermal conversion efficiency due to PDA under NIR irradiation. The heat generated by NIR stimulation leads to the swelling of the hydrogel network and induces drug release. Furthermore, NIR irradiation enhances the CHT effect by increasing the membrane permeability of the drug and facilitating PTT. DOX was loaded onto Fe_3_O_4_@PDA nanocubes through van der Waals forces, including π–π stacking and hydrogen bonding, forming Fe_3_O_4_@PDA@DOX. The synthesized Fe_3_O_4_@PDA@DOX had a ζ-potential of 20.9 mV. The CS-based hydrogel was synthesized through hydrogen bonding and ionic crosslinking between the positively charged amine group (-NH_2_) of CS under acidic conditions and the negatively charged phosphate group (PO_4_^3−^) of β-sodium glycerophosphate (GP). During the hydrogel process, hydroxypropyl cellulose (HPC) was added to reduce the gelation time, enhance the uniformity of the porous structure, and increase the water absorption and swelling ratio of the hydrogel. The resulting CS/HPC/GP hydrogels were then synthesized as a result of this process. Fe_3_O_4_@PDA@DOX was added during the CS/HPC/GP hydrogel process to synthesize Gel@DOX. The temperature of the Gel@DOX increased under NIR irradiation due to the uniformly distributed Fe_3_O_4_@PDA within the hydrogel. The increased temperature induced the release of loaded DOX by decreasing the hydrogen bonding between the nanoparticles and disrupting the crosslinking structure of the hydrogel. DOX dissociation is induced in acidic tumor environments; therefore, drug release was accelerated by the pH of the tumor environment and the NIR irradiation. To confirm its therapeutic effect, Gel@DOX was administered to A549 and HeLa cells. The Gel@DOX-treated group showed 69.0% and 51.1% cell viability in the absence of NIR irradiation. Under NIR irradiation, A549 and HeLa cell viabilities decreased to 10.8% and 8.3%, respectively. This result implies that combination therapy exhibited a stronger therapeutic effect than monotherapy alone, and the study presented a dual-stimuli-responsive gel–nanoparticle hybridization system for cancer therapy.

Manhas et al. developed a CS-based hydrogel system consisting of thermoplastic polyurethane (TPU)-poly(lactic-co-glycolic acid) (PLGA) nanoparticles and a CS hydrogel (Figure 6a) [111]. The TPU-PLGA exhibited high biocompatibility, enhanced the mechanical properties of the hydrogels, and facilitated controlled drug release. The TPU-PLGA nanoparticles were synthesized using a nanoprecipitation method. DOX was loaded during the nanoparticle synthesis, resulting in the formation of DOX-loaded TPU-PLGA nanoparticles. DOX-loaded TPU-PLGA was incorporated into the CS hydrogel to form an NP-based hydrogel system (CTP-DOX NPs). When the pH of the chitosan solution was adjusted above 9, the amine groups in chitosan were deprotonated, leading to hydrogel formation. During this process, hydrogen bonding and ionic interactions between chitosan chains increased, forming a 3D network structure. The CTP-DOX NPs system formed a double-enclosed structure, in which the drug was protected within both the hydrogel matrix and the nanoparticles. In the tumor microenvironment (pH 4.8), the amine group (-NH_2_) of CS became positively charged, increasing intermolecular repulsion. This accelerated the hydrogel swelling and degradation, promoting the release of the loaded drug (Figure 6b). TPU-PLGA-DOX and CTP-DOX NPs hydrogels were administered to HeLa cells to compare their therapeutic effects. The CTP-DOX NP hydrogel-treated group demonstrated a greater therapeutic effect than the TPU-PLGA-DOX nanoparticle-treated group. Furthermore, the CTP-DOX NP hydrogel exhibited a sustained drug release profile with decreased cell toxicity (Figure 6c). That study demonstrates that the hybridization of TPU-PLGA nanoparticles and CS hydrogel can facilitate pH-responsive drug release and enhance the therapeutic effect in cancer.

### 2.4. Active Cancer Targeting

Nanoparticle-incorporated gels can be injected directly into tumor tissue, inducing tissue adhesion and facilitating a sustained therapeutic effect within the tumor microenvironment [112]. Recent studies and reports have shown that hybrid systems not only improve the colloidal stability and biocompatibility of nanoparticles but can also provide active cancer-targeting functions [113,114]. Active targeting can be achieved through receptor-mediated interactions between nanoparticles and cancer cells by constructing a gel with materials that interact with tumor-specific receptors, or by incorporating cancer-targeting molecules into the gel structure [115,116]. In this section, a gel nanoparticle hybrid system is described, in which the gel endows the nanoparticles with active cancer-targeting abilities.

Hou et al. developed a composite nanoparticle-based drug delivery system (FAMP) [117]. FAMP was synthesized by modifying AuNPs and manganese-4-carboxyphenyl porphyrin (Mn-TCPP) on the surface of Fe_3_O_4_ nanoparticles and then coating them with polydopamine (PDA) to enhance stability. Synthesized FAMP was incorporated into a hydrogel consisting of polyvinyl alcohol–carboxylic acid (PVA-COOH) and folic acid-modified polyethylene glycol (FA-PEG-NH_2_), to form the PCF-FAMP system. FAMP facilitates various theranostic applications. The Fe_3_O_4_ properties not only enable MRI imaging but also contribute to cancer treatment by generating ROS through the Fenton reaction. Au generates heat energy under NIR irradiation and facilitates PTT. Mn-TCPP facilitated PDT by generating singlet oxygen. PDA increased the stability of nanoparticles and maximized the therapeutic effect by enhancing the efficiency of photothermal conversion during NIR irradiation. During the hydrogel process, FAMP was added to form the FAMP-incorporated hydrogel (PCF-FAMP), which was synthesized through chemical crosslinking between PVA-COOH and FA-PEG-NH_2_. Folic acid, which was modified on the hydrogel, facilitates selective accumulation of the drug in cancer cells by interacting with folate receptors and reducing drug accumulation in normal cells. Under NIR irradiation, the hydrogel temperature increased, inducing the welling of the hydrogel network. In addition, the hydrogel was shown to be degraded by the acidic tumor environment, thereby promoting drug release. PCF-FAMP not only treats tumor cells through PTT and PDT but also induces ICD and enhances immune responses through combined immune checkpoint blockade (ICB). In vitro experiments showed that the therapeutic effect of PCF-FAMP in HeLa cells increased under NIR irradiation. PCF-FAMP was associated with high levels of cell viability in H8 normal cells. PCF-FAMP was administered to tumor-bearing mice to confirm its therapeutic effects. The PCF-FAMP-administered group demonstrated a significant decrease in tumor size and the method facilitated real-time monitoring through MRI and fluorescence imaging, and the study presented a cancer-targeting gel–nanoparticle hybrid system using a folic acid-modified hydrogel.

Chen et al. developed a cancer-targeting gel–nanoparticle hybrid system consisting of ZIF-8, hyaluronic acid (HA), glucose oxidase (GOx), and an ROS-sensitive crosslinker, acetone-[bis-(2-amino-ethyl)-dithioacetal] (TK) (Figure 7a) [118]. GOx oxidizes glucose to produce gluconic acid and hydrogen peroxide (H_2_O_2_). Gluconic acid promotes the decomposition of the ZIF-8 core structure, while H_2_O_2_ increases ROS concentration, causing oxidative stress and inducing apoptosis in cancer cells. In addition, GOx inhibits cell proliferation by consuming glucose. GOx was covalently bonded with HA through an EDC/NHS reaction to form GOx-HA. Hyaluronic acid is a polymer that occurs naturally in the body and exhibits excellent biocompatibility and biodegradability. Additionally, HA selectively interacts with the CD44 receptor, which is overexpressed on the surface of cancer cells, facilitating active targeting of the system. DOX-loaded ZIF-8 (ZIF-DOX) was modified with GOx-HA, and TK was added to produce a hydrogel (ZIF@HAgel-GOx) that formed a gel hybrid on the ZIF-DOX surface. The amine group (-NH_2_) of TK and the carboxyl group (-COOH) of HA formed an amide bond, leading to the formation of a 3D network structure which subsequently formed a hydrogel. As ZIF@HA-GOx transitioned to ZIF@HAgel-GOx, the ζ-potential changed from 32.3 mV to −22.6 mV. This change was attributed to the utilization of HA’s carboxylic acid groups in gel formation, leading to a decrease in the ζ-potential. The TK cross-linker facilitated HA gel decomposition in response to ROS stimuli, promoting glucose release. In addition, ZIF-8 was exposed to an acidic environment, which induced release of DOX. The ZIF-DOX@HA gel and ZIF-DOX were administered to HepG2 cells to confirm their cancer-targeting effects. The ZIF-DOX@HA gel-administered group demonstrated higher toxicity than the ZIF-DOX-administered group (Figure 7b). This result implied that the HA gel, which was hybridized with nanoparticles, increased the uptake of nanoparticles by interacting with the CD44 receptor. In addition, ZIF@HAgel-GOx significantly increased ROS generation, reducing the HepG2 cell survival rate to approximately 10% (Figure 7c). That study showed the possibility of developing a system capable of active cancer targeting through the hybridization of HA gel and ZIF-8.

Lin et al. developed a cancer-targeting system consisting of hyaluronic acid (HA) and graphene quantum dots (GQDs) [119]. Polyethyleneimine (PEI)-modified GQD (GPI) possesses a positive charge and can load DOX through π–π interactions. Pyrene-modified HA (HA-PB) encapsulated the hydrophobic drug TAK-632 and facilitated cancer targeting by interacting with the CD44 receptor. High-molecular-weight HA exhibited hydrogel-like properties that enhanced drug loading efficiency and enabled controlled drug release. HA-PB and GPI formed a hybrid system (HANPs (TAK)/GPI (DOX) through electrostatic interactions. The synthesized HANPs (TAK)/GPI (DOX) had a size of 700 nm and a ζ-potential of −43.2 mV. Specifically, GPI enhanced the cellular uptake, whereas HA facilitated sustained drug release and CD44-mediated cancer targeting. This system demonstrated high cytotoxicity in HCT116 cells in vitro. Furthermore, in vivo experiments showed that this system effectively inhibited tumor growth. In particular, HANPs (TAK)/GPI (DOX) combination treatment demonstrated an enhanced tumor-suppressive effect compared with a single drug delivery system, while maintaining low toxicity and high biocompatibility. The study results suggest that the hydrogel-like properties of HA can enhance drug delivery efficiency and improve cancer targeting, thereby facilitating the development of new gel–nanoparticle hybrid strategies for cancer treatment.

### 2.5. Role of Nanoparticle Type in Hybrid Gel–Nanoparticle Systems

Nanoparticles in the hybrid system determine the function and performance of the hybrid system. Various nanoparticles may share similar properties, but each has its own unique physical or chemical properties for cancer therapy. In this section, we briefly review the specific properties of nanoparticles based on their type.

Lipid-based nanoparticles (LNPs) consist of a lipid monolayer or bilayer with properties similar to those of the cell membrane [120,121]. Thus, LNPs have high affinity for the lipid layer of the cell membrane, enhancing cell penetration through the endocytosis mechanism. Furthermore, hydrophilic and hydrophobic drugs can both be loaded into LNPs: hydrophilic drugs can be encapsulated in the cores of the LNPs, while hydrophobic drugs can be incorporated into the lipid layer. Polymers such as poly(lactic-co-glycolic acid) (PLGA), polyethylene glycol (PEG), and polyvinyl alcohol (PVA)-based nanoparticles facilitate sustained drug release by controlling cross-linking or polymeric network formation [122,123]. In addition, PLGA, PEG, and PVA have low toxicity and high biocompatibility because these polymers naturally degrade in the body. Metal particles such as gold nanoparticles (AuNPs), silver nanoparticles (AgNPs), iron oxide nanoparticles (Fe_3_O_4_), and MOFs facilitate PDT, PTT, and imaging through responses to external stimuli [43,124]. This not only facilitates complex treatment, but the generated heat or singlet oxygen (^1^O_2_) induces the gel–sol transition or degradation of the hydrogel, allowing controlled drug release.

## 3. Conclusions

Hybrid systems of gel and nanoparticles for cancer therapy have been developed, and several patents have been filed in this field [125,126,127,128]. However, their clinical applications are still in their infancy. To reduce the gap between research and clinical applications, additional research and regulatory approvals are needed. Advanced synthesis techniques such as microfluidics, emulsion-based methods, and 3D bioprinting should be utilized to ensure the reproducibility and scalability of gel–nanoparticle hybrid systems. Synthesized system should be evaluated for nanoparticle size and dispersion, shape and structural characteristics, as well as viscoelasticity and mechanical stability through dynamic light scattering, transmission electron microscopy, and rheological testing, respectively. Additionally, the body clearance mechanism and excretion routes must be studied by analyzing the degradation speed of nanoparticles to ensure biostability and safety, because certain nanoparticles can accumulate in certain organs of the body such as the liver, kidneys, and brain. For this purpose, toxicity and inflammatory responses can be evaluated through in vitro and in vivo experiments, and the potential for side effects due to long-term exposure should be assessed through preclinical and clinical studies. Multidrug resistance (MDR) is one of the major causes of cancer treatment failure, and various strategies are required to overcome it. Combined treatment with drugs and MDR transporter inhibitors can be a solution for overcoming MDR. In addition, photothermal and photodynamic therapy can interfere with the function of MDR transporters, presenting the possibility of overcoming MDR when applied together with drugs. Lastly, for the commercialization of developed systems, preclinical and clinical trials are essential to comply with regulatory agency guidelines in every country, and securing raw materials at a competitive price must also be considered. Through this process, hybrid systems of gels and nanoparticles can become established as effective alternatives for cancer treatment, while advancements in technological, clinical, and industrial aspects should continue. Table 1 summarizes the research on the hybrid systems described in this review.

## Figures and Tables

**Figure 1 gels-11-00170-f001:**
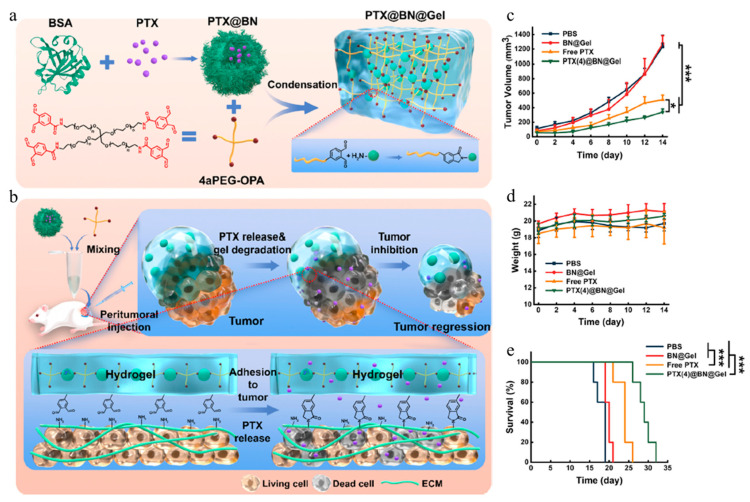
Schematic illustration of (**a**) the preparation of PTX-loaded nanoparticle-incorporated hydrogel and (**b**) the tissue-adhesive hydrogel as a local drug delivery system for effective antitumor chemotherapy in the 4T1 model in vivo. In vivo experimental data of PTX-loaded nanoparticle-incorporated hydrogel. (**c**) Average tumor volume curves with time (n = 5); (**d**) bodyweights over time (n = 5); (**e**) survival curve in different groups (n = 6). (* *p* < 0.05, and *** *p* < 0.001). Reproduced with permission [83]. Copyright (2024): American Chemical Society.

**Figure 2 gels-11-00170-f002:**
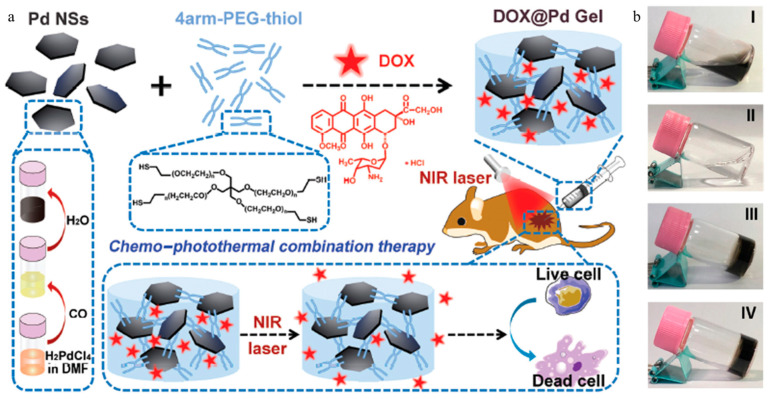
Schematic illustration of (**a**) preparation of DOX@Pd gel and the anticancer capacity of combined chemo-photothermal therapy. (**b**) Characterization of DOX@Pd Gel. Photographs of Pd NS aqueous dispersion (**I**), 4arm-PEG-thiol solution (**II**), Pd gel (**III**) and DOX@Pd gel (**IV**). Reproduced with permission [87]. Copyright (2020): The Royal Society of Chemistry.

**Figure 3 gels-11-00170-f003:**
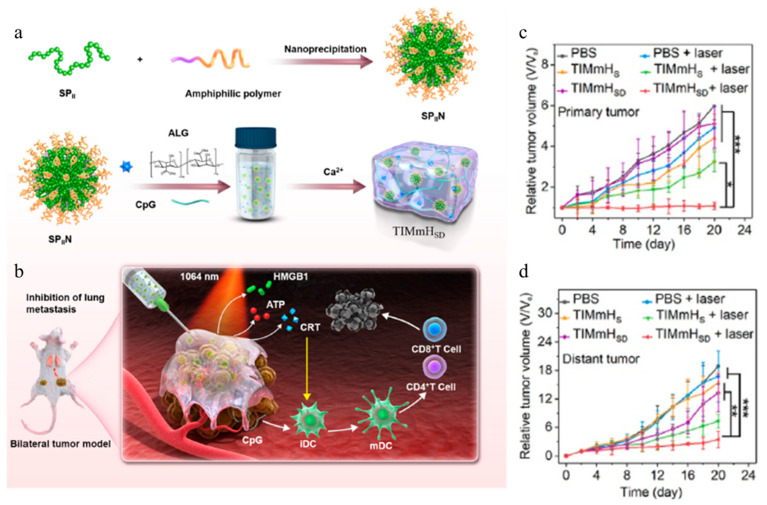
Schematic illustration of (**a**) the preparation process of injectable TIMmH_SD_ hydrogel and (**b**) the mechanism of TIMmH_SD_-mediated thermal immunotherapy. In vivo synergistic therapeutic effect of TIMmH_SD_ hydrogel on primary tumor volume (**c**) and distant tumor volume (**d**) in mice following systemic treatment with PBS, TIMmH_S_, or TIMmH_SD_ via local injection with and without 1064 nm laser irradiation (1 W/cm^2^, 5 min). (* *p* < 0.05, ** *p* < 0.01, and *** *p* < 0.001). Reproduced with permission [97]. Copyright (2022): Elsevier.

**Figure 4 gels-11-00170-f004:**
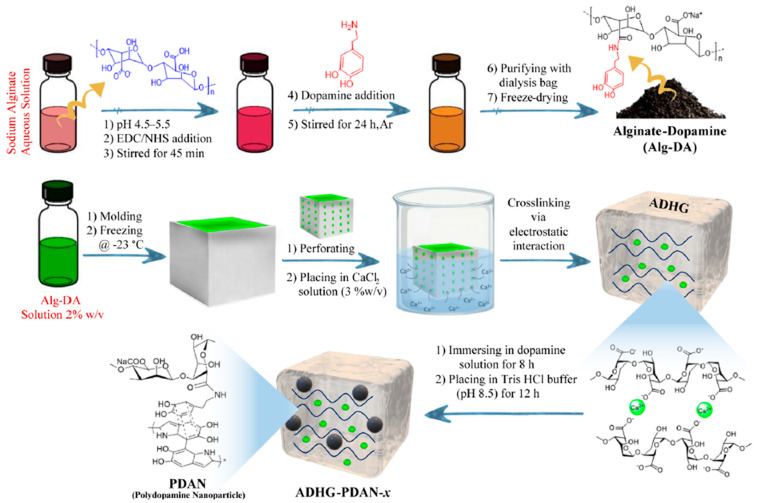
ADHG hydrogels are initially crosslinked via electrostatic interactions between calcium cations and the carboxylate groups of Alg-DA. Their stability is further enhanced by incorporating polydopamine nanoparticles, which induce hydrogen bonding and π–π stacking interactions with the catechol groups of Alg-DA chains. Reproduced with permission [98]. Copyright (2025): American Chemical Society.

**Figure 5 gels-11-00170-f005:**
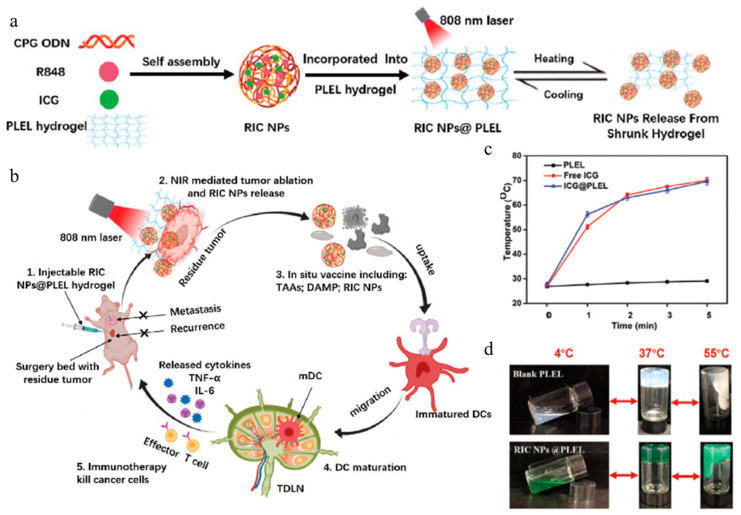
Schematic illustration of (**a**) the preparation process of RIC NPs@PLEL hydrogels and (**b**) photothermal immune therapy to prevent post-surgery tumor reoccurrence. (**c**) Temperature changes of PLEL hydrogel under NIR laser irradiation. (**d**) Reversible sol–gel transformation photos of PLEL hydrogels and RIC NPs@PLEL hydrogels. Reproduced with permission [108]. Copyright (2020): John Wiley & Sons.

**Figure 6 gels-11-00170-f006:**
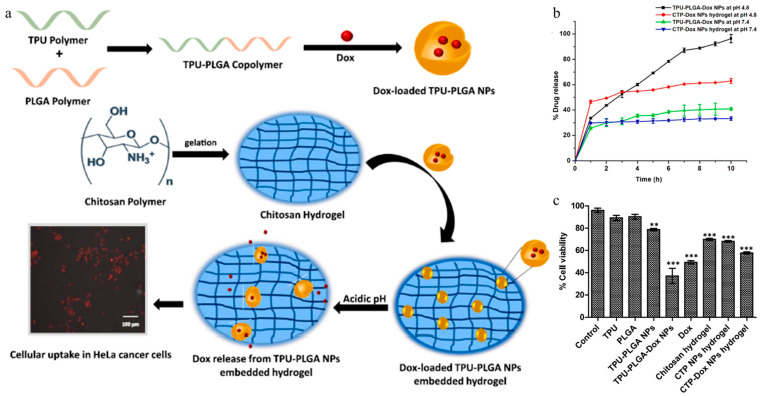
Schematic illustration of (**a**) the fabricated DOX-loaded TPU-PLGA NPs incorporated into chitosan hydrogel for anticancer drug delivery in HeLa cancer cells. (**b**) In vitro release profile of TPU-PLGA-DOX NPs and CTP-DOX NP hydrogels. (**c**) A cell viability assay was performed to analyze the effect of the nano-formulations on HeLa cells using 8.5 μg/mL (IC_50_) DOX. Dunnett’s test was performed for statistical analysis (** *p* < 0.05, *** *p* < 0.001). Reproduced with permission [111]. Copyright (2023): Elsevier.

**Figure 7 gels-11-00170-f007:**
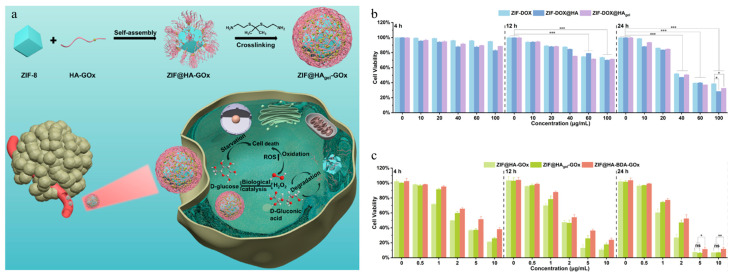
(**a**) Schematic illustration of the synthesis and intracellular action mechanism of ZIF@HAgel-GOx. (**b**) Cell viability plotted against concentration of ZIF-DOX, ZIF-DOX@HA, and ZIF-DOX@HA gel, which were co-incubated with HepG2 cells for 4, 12, and 24 h. (**c**) Cell viability plotted against concentration of ZIF@HA-GOx, ZIF@HAgel-GOx, and ZIF@HA-BDA-GOx, which were co-incubated with HepG2 cells for 4, 12, and 24 h. *** *p* < 0.001, ** *p* < 0.01, and * *p* < 0.05; ns: not significant. Reproduced with permission [118]. Copyright (2024): American Chemical Society.

**Table 1 gels-11-00170-t001:** Summary of current studies on gel–nanoparticle hybrid systems.

Gel Type	Fabrication Method	Gelling Agent	Nanoparticle	Therapy Method	Drug	Ref
Hydrogel	Desolvation	BSA, branched PEG	BSA	Chemotherapy	PTX	[83]
Lipogel	Buffer switch method	2-Naph-L-Phe-Z-ΔPhe-OH	MNP, MS-AuNR	Chemotherapy, photothermal therapy, magnetic hyperthermia	DOX, PTX	[84]
Colloidal gel	Coacervation andthermal crosslinking	C10-ApGltn	SPIONs	Magnetic hyperthermia	N/A	[85]
HA-BP-basedhydrogel	Dynamic coordination bonds	HA-BP, MOF	MOF	Chemotherapy	DOX,	[86]
Pd NS-knottedhydrogel	Dynamic Pd–S bonds	Pd NS, 4arm-PEG-thiol	Pd NS	Chemotherapy, photothermal therapy	DOX,	[87]
Alginate-basedhydrogel	Coordination	Alginate	SPIIN	Photothermal therapy, immunotherapy	CpG ODNs	[97]
Alginate-basedhydrogel	Electrostatic interaction	Alg-DA	PDA NP	Photothermal therapy	N/A	[98]
GelMA basedhydrogel	(Not specified in the paper)	GelMA	ZIF-8	Chemotherapy	DOX	[78]
Alginate/gelatin-based hydrogel, Chitosan hydrogel	Schiff base reaction	CS, OAL, Gelatin, β-GP	AuNP	Chemotherapy	DOX	[99]
PLEL hydrogel	3D micelle network	PLEL	RIC NP	Photothermal therapy, immunotherapy	ICG, R848, CpG ODNs	[108]
Chitosan–puerarin hydrogel	Grinding methodwith acetic acid	CS-puerarin, AuNRs	AuNRs	Photothermal therapy, gene-targeted therapy	DC_AC50, Puerarin	[109]
Chitosan–HPChydrogel	Temperature-inducedgelation	CS, GP, HPC	Fe_3_O_4_@PDA	Photothermal therapy, chemotherapy	Dox	[110]
Chitosan hydrogel	pH-induced gelation	CS, NaOH	TPU-PLGA-Dox	Chemotherapy	Dox	[111]
PVA/PEGhydrogel	Ionic crosslinking, hydrogen bonding	PVA, PEG	Fe_3_O_4_@Au/Mn-TCPP	Photothermal therapy, photodynamic therapy,immunotherapy	N/A	[117]
Hyaluronic acid hydrogel	Amide bond-mediated crosslinking	HA, TK	ZIF@HAgel-GOx	Starvation therapy, oxidative therapy	Dox, GOX	[118]
High molecular weight HA	Desolvation	HA	CQD-PEI /HA-PB	Dual-drug therapy(chemotherapy)	Dox, TAK-632	[119]

Notes: GelMA, gelatin methacryloyl; PLEL, PDLLA-PEG-PDLLA; HPC, hydroxypropyl cellulose; PVA, polyvinyl alcohol–carboxylic acid; PEG, polyethylene glycol; C10-ApGltn, decyl-group-modified Alaskan pollock gelatin; HA-BP, bisphosphonate-functionalized hyaluronic acid; 4arm-PEG-thiol, thiol-terminated four-arm polyethylene glycol; Alg-DA, dopamine-grafted alginate; CS, chitosan; OAL, oxidized alginate; β-GP, β-glycerophosphate; GP, β-sodium glycerophosphate; HPC, hydroxypropyl cellulose; GP, β-sodium glycerophosphate; HA, hyaluronic acid; TK, acetone-[bis-(2-amino-ethyl)-dithioacetal]; BSA, bovine serum albumin; MNP, manganese-based magnetic nanoparticles; MS-AuNR, mesoporous silica-coated gold nanorods; SPION, superparamagnetic iron oxide nanoparticles; MOF, metal–organic frameworks; Pd NS, palladium nanosheets; SPIIN, semiconducting polymer nanoparticle; PDA NP, polydopamine NPs; ZIF-8, zeolitic imidazolate framework-8; AuNP, gold nanoparticle; RIC NPs, indocyanine green, resiquimod, and cytosine–phosphorothioate–guanine oligodeoxynucleotide nanoparticles; GQDs, graphene quantum dots; HA-PB, HA-1-pyrenebutyric; PTX, paclitaxel; Dox, doxorubicin; CpG ODN, cytosine–phosphorothioate–guanine oligodeoxynucleotide; ICG, indocyanine green; R848, resiquimod; GOx; glucose oxidase.

## Data Availability

No new data were created or analyzed in this study.

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
