# Peer review of "Hybrid Systems of Gels and Nanoparticles for Cancer Therapy: Advances in Multifunctional Therapeutic Platforms"

_gels, 2025, doi:10.3390/gels11030170_

Round 1

Reviewer 1 Report

Comments and Suggestions for Authors
  • The article is well-researched and informative, providing valuable insights into the hybrid gel-nanoparticle approach for cancer therapy.
  • A thorough proofreading for grammar and citation consistency would enhance readability.
  • Adding comparative data and clinical perspectives would strengthen its impact.

And there're some questions to improving this article :

  • Reproducibility and Scalability: How can these systems be standardized for large-scale production while maintaining efficacy?
  • Biostability and Safety: What are the long-term effects of gel-nanoparticle systems on the human body?
  • Overcoming Multidrug Resistance (MDR): How can hybrid systems be optimized to prevent MDR in cancer cells?
  • Clinical Translation: What regulatory and cost-related barriers exist in bringing these advanced therapeutic platforms to market?

Author Response

Comments to the Author
The article is well-researched and informative, providing valuable insights into the hybrid gel-nanoparticle approach for cancer therapy. A thorough proofreading for grammar and citation consistency would enhance readability. Adding comparative data and clinical perspectives would strengthen its impact. And there're some questions to improving this article.

Comment:

1) Reproducibility and Scalability: How can these systems be standardized for large-scale production while maintaining efficacy?

Answer:

Thank you for the valuable comments. The manuscript is corrected as follows.

 “Advanced synthesis techniques such as microfluidics, emulsion-based methods, and 3D bioprinting should be utilized to ensure the reproducibility and scalability of gel-nanoparticle hybrid systems. The synthesized system is evaluated for nanoparticle size and dispersion, shape and structural characteristics, as well as viscoelasticity and mechanical stability through dynamic light scattering, transmission electron microscopy, and rheological testing, respectively.” (Page 15, line 573-578)

Comment:

2) Biostability and Safety: What are the long-term effects of gel-nanoparticle systems on the human body?

Answer:

Thank you for the valuable comments. The manuscript is corrected as follows.

“Additionally, the body clearance mechanism and excretion routes must be studied by analyzing the degradation speed of nanoparticles to ensure biostability and safety. Because certain nanoparticles can accumulate in certain organs of the body, such as the liver, kidneys and brain. For this purpose, toxicity and inflammatory responses are evaluated through in-vitro and in-vivo experiments, and the potential for side effects due to long-term exposure should be assessed through preclinical and clinical studies.” (Page 15, line 579-584)

Comment:

3) Overcoming Multidrug Resistance (MDR): How can hybrid systems be optimized to prevent MDR in cancer cells?

Answer:

Thank you for the valuable comments. The manuscript is corrected as follows.

“Multidrug resistance (MDR) is one of the major causes of cancer treatment failure, and various strategies are required to overcome it. Combined treatment with drugs and MDR transporter inhibitors can be a solution for overcoming MDR. In addition, photothermal and photodynamic therapy can interfere with the function of MDR transporters, presenting the possibility of overcoming MDR when applied together with drugs.” (Page 15, line 584-589)

Comment:

4) Clinical Translation: What regulatory and cost-related barriers exist in bringing these advanced therapeutic platforms to market?

Thank you for the valuable comments. The manuscript is corrected as follows.

 “Lastly, for the commercialization of the developed system, preclinical and clinical trials are essential to comply with regulatory agency guidelines in each country, and securing the price competitiveness of raw materials must also be considered. Through this process, the hybrid system of gel and nanoparticle can establish itself as an effective alternative for cancer treatment, while advancements in technological, clinical, and industrial aspects should continue.” (Page 15, line 589-594)

Reviewer 2 Report

Comments and Suggestions for Authors

Kibeom Kim has presented a review based on the application of a gel system carrying metal nanoparticles for cancer treatment management. The review is very informative. However, some points need to be addressed and improved before its publication.

  1. The author should add a separate segment explaining the usefulness of the lipid, polymer, and hybrid polymer lipid systems along with metal NPs and their gelling systems in cancer treatment.
  2. The author should give detailed information about the type of gelling agent applied, how natural or herbal gels could be beneficial, and the method of cross-linking.
  3. The author should include some comparative studies on the surface changes of the NPs after their incorporation into the gels.
  4. The author should provide updated information about patents filed and their clinical translation status.

Good Luck!

Author Response

Comments to the Author
Kibeom Kim has presented a review based on the application of a gel system carrying metal nanoparticles for cancer treatment management. The review is very informative. However, some points need to be addressed and improved before its publication.

Comment:

1) The author should add a separate segment explaining the usefulness of the lipid, polymer, and hybrid polymer lipid systems along with metal NPs and their gelling systems in cancer treatment.

Answer:

Thank you for the valuable comments.

In the gel and nanoparticle hybrid system, research has generally focused on the advantages of hybridization between the gel and nanoparticles. As a result, there are not many papers that explain the unique role of nanoparticles, making it difficult to identify commonalities among them. Therefore, a section has been added to the manuscript, as shown below, to explain the unique properties of each nanoparticle and the necessity of understanding nanoparticle properties in the hybrid system.

2.5. Role of Nanoparticle type in Hybrid system of Gel and Nanoparticle.

Nanoparticles in the hybrid system determine the function and performance of the hybrid system. Various nanoparticles may share similar properties, but each has its own unique physical or chemical properties for cancer therapy. In this section, we briefly re-view the specific properties of nanoparticles based on their type.

Lipid-based nanoparticles (LNPs) consist of a lipid monolayer or bilayer, which have properties similar to those of the cell membrane [120, 121]. Thus, LNPs have a high affinity for the lipid layer of the cell membrane, enhancing cell penetration through the endocytosis mechanism. Furthermore, both hydrophilic and hydrophobic drugs can be loaded into LNPs: hydrophilic drugs can be encapsulated in the core of LNPs, while hydrophobic drugs can be incorporated into the lipid layer of LNPs. Polymers such as poly(lactic-co-glycolic acid) (PLGA), polyethylene glycol (PEG), and polyvinyl alcohol (PVA)-based nanoparticles facilitate sustained drug release by controlling cross-linking or polymeric network formation [122, 123]. In addition, PLGA, PEG, and PVA have low toxicity and high biocompatibility because these polymers naturally degrade in the body. Metal particles such as gold nanoparticles (AuNPs), silver nanoparticles (AgNPs), iron oxide nanoparticles (Fe3O4), and MOFs facilitate PDT, PTT, and imaging through responses to external stimuli [124, 125]. This not only facilitates complex treatment, but the generated heat or singlet oxygen (1O2) induces the gel-sol transition or degradation of the hydrogel, allowing for controlled drug release.” (Page 15, line 549-568)

“120.     Sheoran, S.; Arora, S.; Samsonraj, R.; Govindaiah, P. ; Vuree, S.,Lipid-based nanoparticles for treatment of cancer. Heliyon 2022,8,e09403.

  1. Waheed, I.; Ali, A.; Tabassum, H.; Khatoon, N.; Lai, W.F. ; Zhou, X.,Lipid-based nanoparticles as drug delivery carriers for cancer therapy. Front Oncol 2024,14,1296091.
  2. Subhan, M.A. ; Muzibur Rahman, M.,Recent Development in Metallic Nanoparticles for Breast Cancer Therapy and Diagnosis. Chem Rec 2022,22,e202100331.
  3. Villalobos Gutierrez, P.T.; Munoz Carrillo, J.L.; Sandoval Salazar, C.; Viveros Paredes, J.M. ; Gutierrez Coronado, O.,Functionalized Metal Nanoparticles in Cancer Therapy. Pharmaceutics 2023,15, 1932.
  4. Bisht, S.; Feldmann, G.; Soni, S.; Ravi, R.; Karikar, C.; Maitra, A. ; Maitra, A.,Polymeric nanoparticle-encapsulated curcumin ("nanocurcumin"): a novel strategy for human cancer therapy. J Nanobiotechnology 2007,5,3.
  5. Masood, F.,Polymeric nanoparticles for targeted drug delivery system for cancer therapy. Mater Sci Eng C Mater Biol Appl 2016,60,569-578.” (Page 23, line 883-895)

Comment:

2) The author should give detailed information about the type of gelling agent applied, how natural or herbal gels could be beneficial, and the method of cross-linking.

Answer:

Thank you for the valuable comments.

The gelling agents and gelation methods used in the referenced papers have been included in the content and additionally organized in a table for clarity. Furthermore, the types and characteristics of the gels used have been explained and added as follows.

“After synthesis, the PTX-loaded nanoparticles (PTX@BN) were mixed with branched polyethylene glycol (PEG) and OPA functional groups, which underent a condensation reaction with the primary amine of BSA to form a PTX@BN-loaded hydrogel (PTX@BN gel) in situ. After synthesis, PTX-loaded nanoparticles (PTX@BN) were mixed with branched PEG containing OPA functional groups. PEG is a synthetic, hydrophilic polymer with high biocompatibility, and it can enhance the biocompatibility of the gel when incorporated into its structure.”

(Page 2, line 79-86)

“Lipogels were synthesized through the buffer switch method using 2-Naph-L-Phe-Z-ΔPhe-OH (compound 1 or C1) and 2-Naph-L-Phe-Z-ΔPhe-FRGDG-OH (compound 2 or C2), with C2 containing cancer targeting peptide sequence (Phe-Arg-Gly-Asp-Gly, FRGDG).” (Page 3, line 109-112)

“Gelatin is a protein-based natural polymer obtained through the hydrolysis of collagen. It has the advantages of high biocompatibility and excellent biodegradability. Additionally, it undergoes a reversible gel-sol transition in water.” (Page 4, line 136-138)

“The synthesized C10-ApGltn formed a coacervate in the solvent, which was transformed into C10MPs of 2.3 ± 1.0 μm by thermal crosslinking. When a mixture of SPIONs and C10MPs was sprayed onto cancer tissue, C10MPs formed a colloidal gel through hydro-phobic interactions under the moist conditions and stably encapsulate dthe SPIONs through coordination bonds.” (Page 4, line 141-145)

“DOX loaded MOF (MOF@DOX) and HA-BP form the HA-BP·MOF@DOX hydrogel via dynamic coordination bond.” (Page 4, line 166-167)

“Pd NS possess a high photothermal conversion efficiency of 52% under 808 nm NIR light and can be easily surface-modified through palladium-sulfur (Pd-S) bonding. This property of the Pd NS facilitated the formation of a hydrogel (Pd Gel) with 4arm-PEG-thiol.” (Page 5, line 184-187)

“Alginate, a natural polysaccharide, exhibits excellent biocompatibility and biodegradabil-ity and is widely used in medicine, drug delivery, and tissue engineering.” (Page 6, line 222-224)

“The carboxylic acid groups of ALG and Ca²⁺ ions in the body form a stable hydrogel through coordination bonding.” (Page 6, line 227-228)

“The synthesized Alg-DA formed the ADHG through electrostatic interactions with Ca2+ ions via crosslinking after being placed in a mold, frozen, and immersed in a CaCl2 solu-tion. The synthesized ADHG was immersed in a dopamine solution to generate PDA NPs inside the gel, induced through dopamine self-oxidation.” (Page 7, line 261-264)

"GelMA is synthesized through the reaction between the lysine residues of gelatin and methacrylic anhydride. The synthesized GelMA forms a gel in the presence of a photoinitiator and under UV irradiation." (Page 8, line 283-286)

“Chitosan, a natural polysaccharide derived from the chitin of crustaceans, is receiving much attention in drug delivery systems due to its high biocompatibility and enzymatic degradability in the body. The surface of AuNPs was modified with CS, which then formed a CS gel through ionic crosslinking between the amine groups (-NH2) of CS and the anionic groups (PO43-) of sodium triphosphate pentabasic (TPP).” (Page 8, line 287-290)

“DOX-loaded CS/AuNPs (CS/AuNPs-DOX) were added to a solution consisting of OAL, gelatin, and beta-glycerophosphate (β-GP). This solution was transformed into the CS/AuNPs-DOX-encapsulated hydrogel (OALGH) through a Schiff-base reaction between the aldehyde (-CHO) groups of OAL and the amine (-NH2) groups of gelatins. β-GP, an ionic crosslinker, controls the hydrogel pH from acidic to neutral and induces gelation by accelerating the sol-to-gel transition.” (Page 8, line 315-321)

"PLEL self-assembles into micelles, which further aggregate to form a 3D micelle network, ultimately leading to hydrogel formation." (Page10, line 357-358)

“Puerarin forms a hydrogel through self-assembly via a rapid heating-cooling process; however, puerarin hydrogel has a rigid nanofiber structure, posing a challenge for direct injection.” (Page11, line 377-379)

“After grinding, the CP@Au@DC_AC50 hydrogel was synthesized through various interac-tions, including hydrogen bonding, π–π interactions, hydrophobic interactions, van der Waals forces, and electrostatic interactions. The AuNRs enhanced the mechanical strength of the hydrogel and facilitated the controlled release of DC_AC50 by inducing PTT and a temperature-responsive gel-sol transition under NIR irradiation.” (Page11, line 384-389)

“The CS-based hydrogel was synthesized through hydrogen bonding and ionic cross-linking between the positively charged amine group (-NH2) of CS under acidic conditions and the negatively charged phosphate group (PO43-⁻) of β-sodium glycerophosphate (GP). During the hydrogel process, hydroxypropyl cellulose (HPC) was added to reduce the ge-lation time, enhance the uniformity of the porous structure, and increase the water ab-sorption and swelling ratio of the hydrogel.” (Page11, line 409-414)

"When the pH of the chitosan solution is adjusted above 9, the amine groups in chitosan are deprotonated, leading to hydrogel formation. During this process, hydrogen bonding and ionic interactions between chitosan chains increase, forming a 3D network structure." (Page12, line 436-439)

“During the hydrogel process, FAMP was added to form the FAMP-incorporated hydrogel (PCF-FAMP), which was synthesized through chemical crosslinking between PVA-COOH and FA-PEG-NH2.” (Page13, line 480-482)

“Hyaluronic acid is a natural polymer that occurs naturally in the body and exhibits excel-lent biocompatibility and biodegradability. Additionally, HA selectively interacts with the CD44 receptor, which is overexpressed on the surface of cancer cells, facilitating active targeting of the system.” (Page13, line 503-506)

“The amine group (-NH2) of TK and the carboxyl group (-COOH) of HA form an amide bond, leading to the formation of a 3D network structure, which subsequently forms a hydrogel.” (Page13, line 508-510)

Gel type

Fabrication method

Gelling agent

Nanoparticle

Therapy method

Drug

Ref

Hydrogel

Desolvation

BSA,

branched PEG

BSA

Chemotherapy

PTX

[83]

Lipogel

Buffer switch method

2-Naph-L-Phe-Z-ΔPhe-OH

MNP,

MS-AuNR

Chemotherapy,
Photothermal therapy,
 Magnetic hyperthermia

DOX, PTX

[84]

Colloidal gel

Coacervation and
thermal crosslinking

C10-ApGltn

SPIONs

Magnetic hyperthermia

N/A

[85]

HA-BP-based
hydrogel

Dynamic coordination bonds

HA-BP, MOF

MOF

Chemotherapy

DOX,

[86]

Pd NS-knotted
hydrogel

Dynamic Pd–S bonds

Pd NS,

4arm-PEG-thiol

Pd NS

Chemotherapy,

photothermal therapy

DOX,

[87]

Alginate-based
hydrogel

Coordination

Alginate

SPIIN

Photothermal therapy,

Immunotherapy

CpG ODNs

[97]

Alginate-based
hydrogel

Electrostatic interaction

Alg-DA

PDA NP

Photothermal therapy

N/A

[98]

GelMA based
hydrogel

(Not specified in the paper)

GelMA

ZIF-8

Chemotherapy      

DOX

[78]

Alginate/gelatin-based hydrogel,
Chitosan hydrogel

Schiff-base reaction

CS, OAL,

Gelatin, β-GP

AuNP

Chemotherapy      

DOX

[99]

PLEL hydrogel

3D micelle network

PLEL

RIC NP

Photothermal therapy,

Immunotherapy

ICG, R848, CpG ODNs

[108]

Chitosan-puerarin hydrogel

Grinding method
with acetic acid

CS-puerarin,

AuNRs

AuNRs

Photothermal therapy,

Gene-targeted therapy

DC_AC50, Puerarin

[109]

Chitosan-HPC
hydrogel

Temperature-induced
gelation

CS, GP, HPC

Fe₃O₄@PDA

Photothermal therapy,

Chemotherapy      

Dox

[110]

Chitosan hydrogel

pH-induced gelation

CS, NaOH

TPU-PLGA-Dox

Chemotherapy

Dox

[111]

PVA /PEG
hydrogel

Ionic crosslinking,

hydrogen bonding

PVA, PEG

Fe₃O₄@Au/Mn-TCPP

Photothermal,

Photodynamic,
Immunotherapy

N/A

[117]

Hyaluronic acid hydrogel

Amide bond-mediated crosslinking

HA, TK

ZIF@HAgel-GOx

Starvation therapy,

oxidative therapy

Dox ,

GOX

[118]

High molecular weight HA

Desolvation

HA

CQD-PEI

/ HA-PB

Dual-drug therapy
(chemotherapy)

Dox,

TAK-632

[119]

Table 1. Summary of current studies on gel-nanoparticle hybrid systems.

1 GelMA, gelatin methacryloyl; PLEL, PDLLA-PEG-PDLLA; HPC, hydroxypropyl cellulose; PVA, polyvinyl alcohol-carboxylic acid; PEG, polyethylene glycol; C10-ApGltn, decyl-group-modified alaska pollock gelatin; HA-BP, bisphosphonate-functionalized hyaluronic acid; 4arm-PEG-thiol, thiol-terminated four-arm polyethylene glycol; Alg-DA, dopamine-grafted alginate; CS, chitosan; OAL, oxidized alginate; β-GP, β-glycerophosphate; GP, β-sodium glycerophosphate; HPC, hydroxypropyl cellulose; GP, β-sodium glycerophosphate; HA, hyaluronic acid; TK, Acetone-[bis-(2-amino-ethyl)-dithioacetal]; BSA, bovine serum albumin; MNP, manganese-based magnetic nanoparticles; MS-AuNR, mesoporous silica-coated gold nanorods; SPION, superparamagnetic iron oxide nanoparticles; MOF, metal-organic frameworks; Pd NS, palladium nanosheets; SPIIN, semiconducting polymer nanoparticles; PDA NP, polydopamine NPs; ZIF-8, zeolitic imidazolate framework-8; AuNP, gold nanoparticle; RIC NPs, Iindocyanine gGreen, rResiquimod, , and  cytosine-phosphorothioate-guanine oligodeoxynucleotides nanoparticle;GQD, graphene quantum dots; HA-PB, HA-1-pyrenebutyric; PTX, paclitaxel; Dox, doxorubicin; CpG ODN, cytosine-phosphorothioate-guanine oligodeoxynucleotides; ICG, indocyanine green; R848, resiquimod; GOx; glucose oxidase

Comment:

3) The author should include some comparative studies on the surface changes of the NPs after their incorporation into the gels.

Answer:

Thank you for the valuable comments. Some reference papers provide ζ-potential values for nanoparticles, but do not report ζ-potential values after gel modification. Additionally, some papers do not mention ζ-potential values for nanoparticles at all. Therefore, in this manuscript, the reported ζ-potential values from each paper are summarized and presented as follows.

“Synthesized nanoparticle size is 120~160 nm and a ζ-potential of approximately -15.2 to 10.9 mV.” (Page 2, line 78-79)

“The hydrodynamic diameter of the liposome is 100 nm, and its ζ-potential is -2 mV.” (Page 3, line 108-109)

“The ζ-potential of the synthesized MOF increased from -10.14 mV to 0.016 mV after DOX encapsulation” (Page 4, line 165-166)

“Shen et al. developed a tumor immunosuppressive microenvironment-modulating hydrogel (TIMmHsd), which consists of semiconducting polymer nanoparticles (SPIIN) with a ζ-potential of -9.04 mv, an immunoadjuvant cytosine-phosphorothioate-guanine oligodeoxynucleotide (CpG ODNs), and a Ca²⁺-responsive alginate hydrogel (ALG) for combined PTT and IMT (Figure 3a) [97].” (Page 6, line 218-222)

“The synthesized ZC-DOX has a size of 102 nm and a ζ-potential of 24.2 mV.” (Page 9, line 287-288)

“The synthesized OALGH has a size of 209 nm and a ζ-potential of 19.2 mV.” (Page 9, line 321-322)

“Synthesized Fe3O4@PDA@DOX ζ-potential is 20.9 mV” (Page 11, line 408-409)

“As ZIF@HA-GOx transitioned to ZIF@HAgel-GOx, the ζ-potential changed from 32.3 mV to -22.6 mV. This change is attributed to the utilization of HA's carboxylic acid groups in gel formation, leading to a decrease in the ζ-potential.” (Page 14, line 510-513)

“The synthesized HANPs (TAK)/GPI (DOX) has a size of 700 nm and a ζ-potential of -43.2 mV.” (Page 14, line 539-540)

Comment:

4) The author should provide updated information about patents filed and their clinical translation status.

Answer:

Thank you for the valuable comments. The manuscript is corrected as follows.

“The hybrid system of gel and nanoparticles for cancer therapy has been significantly developed, and several patents have been filed in this field [126-129]. However, its clinical applications are still in their infancy. To reduce the gap between research and clinical applications, additional research and regulatory approvals are needed. Advanced synthesis techniques such as microfluidics, emulsion-based methods, and 3D bioprinting should be utilized to ensure the reproducibility and scalability of gel-nanoparticle hybrid systems. The synthesized system is evaluated for nanoparticle size and dispersion, shape and structural characteristics, as well as viscoelasticity and mechanical stability through dynamic light scattering, transmission electron microscopy, and rheological testing, respectively. Additionally, the body clearance mechanism and excretion routes must be studied by analyzing the degradation speed of nanoparticles to ensure biostability and safety. Because certain nanoparticles can accumulate in certain organs of the body, such as the liver, kidneys and brain. For this purpose, toxicity and inflammatory responses are evaluated through in-vitro and in-vivo experiments, and the potential for side effects due to long-term exposure should be assessed through preclinical and clinical studies. Multi-drug resistance (MDR) is one of the major causes of cancer treatment failure, and various strategies are required to overcome it. Combined treatment with drugs and MDR trans-porter inhibitors can be a solution for overcoming MDR. In addition, photothermal and photodynamic therapy can interfere with the function of MDR transporters, presenting the possibility of overcoming MDR when applied together with drugs. Lastly, for the commercialization of the developed system, preclinical and clinical trials are essential to comply with regulatory agency guidelines in each country, and securing the price competitiveness of raw materials must also be considered. Through this process, the hybrid system of gel and nanoparticle can establish itself as an effective alternative for cancer treatment, while advancements in technological, clinical, and industrial aspects should continue.” (Page 15, line 570-594)

Reviewer 3 Report

Comments and Suggestions for Authors

The manuscript provides a comprehensive review of the recent advancements in hybrid systems combining gels and nanoparticles for cancer therapy. The authors have done a commendable job in summarizing the various strategies employed to enhance drug delivery, improve biocompatibility, and achieve controlled drug release using these hybrid systems. Below, I have provided some suggestions that could help to improve this study.

  • The conclusion section is somewhat brief and does not fully summarize the key findings or provide a clear direction for future research.

  • Some of the references are quite old (e.g., from 2011-2015). The authors should ensure that they are citing the most recent and relevant studies, especially in a rapidly evolving field like nanomedicine. My suggestions are listed below;

https://doi.org/10.3390/cells13110908

https://doi.org/10.1016/j.eurpolymj.2024.113400 

https://doi.org/10.3390/mi14010208

https://doi.org/10.1016/j.jconrel.2015.09.029

doi: 10.32604/or.2024.053069

Author Response

Comments to the Author
The manuscript provides a comprehensive review of the recent advancements in hybrid systems combining gels and nanoparticles for cancer therapy. The authors have done a commendable job in summarizing the various strategies employed to enhance drug delivery, improve biocompatibility, and achieve controlled drug release using these hybrid systems. Below, I have provided some suggestions that could help to improve this study.

Comment:

1) The conclusion section is somewhat brief and does not fully summarize the key findings or provide a clear direction for future research.

Answer:

Thank you for the valuable comments. The manuscript is corrected as follows.

“The hybrid system of gel and nanoparticles for cancer therapy has been significantly developed, and several patents have been filed in this field [126-129]. However, its clinical applications are still in their infancy. To reduce the gap between research and clinical applications, additional research and regulatory approvals are needed. Advanced synthesis techniques such as microfluidics, emulsion-based methods, and 3D bioprinting should be utilized to ensure the reproducibility and scalability of gel-nanoparticle hybrid systems. The synthesized system is evaluated for nanoparticle size and dispersion, shape and structural characteristics, as well as viscoelasticity and mechanical stability through dynamic light scattering, transmission electron microscopy, and rheological testing, respectively. Additionally, the body clearance mechanism and excretion routes must be studied by analyzing the degradation speed of nanoparticles to ensure biostability and safety. Because certain nanoparticles can accumulate in certain organs of the body, such as the liver, kidneys and brain. For this purpose, toxicity and inflammatory responses are evaluated through in-vitro and in-vivo experiments, and the potential for side effects due to long-term exposure should be assessed through preclinical and clinical studies. Multi-drug resistance (MDR) is one of the major causes of cancer treatment failure, and various strategies are required to overcome it. Combined treatment with drugs and MDR trans-porter inhibitors can be a solution for overcoming MDR. In addition, photothermal and photodynamic therapy can interfere with the function of MDR transporters, presenting the possibility of overcoming MDR when applied together with drugs. Lastly, for the commercialization of the developed system, preclinical and clinical trials are essential to comply with regulatory agency guidelines in each country, and securing the price competitiveness of raw materials must also be considered. Through this process, the hybrid system of gel and nanoparticle can establish itself as an effective alternative for cancer treatment, while advancements in technological, clinical, and industrial aspects should continue.” (Page 15, line 570-594)

Comment:

2) Some of the references are quite old (e.g., from 2011-2015). The authors should ensure that they are citing the most recent and relevant studies, especially in a rapidly evolving field like nanomedicine. My suggestions are listed below;

https://doi.org/10.3390/cells13110908

https://doi.org/10.1016/j.eurpolymj.2024.113400

https://doi.org/10.3390/mi14010208

https://doi.org/10.1016/j.jconrel.2015.09.029

https://doi.org/10.32604/or.2024.053069

Answer:

Thank you for the valuable comments. The reference has been changed as follows.

“38. He, C.; Lu, J. ; Lin, W.,Hybrid nanoparticles for combination therapy of cancer. J Control Release 2015,219,224-236.” (Page 19, line 697)

“56. Vargas-Molinero, H.Y.; Serrano-Medina, A.; Palomino-Vizcaino, K.; López-Maldonado, E.A.; Villarreal-Gómez, L.J.; Pérez-González, G.L.; Cornejo-Bravo, J.M.,Hybrid systems of nanofibers and polymeric nanoparticles for biological application and delivery systems. Micromachines 2023,14,208.” (Page 20, line 736-738)

“57. Soleymani, S.; Doroudian, M.; Soezi, M.; Beladi, A.; Asgari, K.; Mobarakshahi, A.; Aghaeipour, A. ; Macloughlin, R.,Engendered nanoparticles for treatment of brain tumors. Oncol Res 2024,33,15.” (Page 20, line 739-740)

“62. Katopodi, T.; Petanidis, S.; Floros, G.; Porpodis, K. ; Kosmidis, C.,Hybrid Nanogel Drug Delivery Systems: Transforming the Tumor Microenvironment through Tumor Tissue Editing. Cells 2024,13,908.” (Page 20, line 749-750)

“63. Farjadian, F.; Mirkiani, S.; Ghasemiyeh, P.; Kafshboran, H.R.; Mehdi-Alamdarlou, S.; Raeisi, A.; Esfandiarinejad, R.; Soleymani, S.; Goshtasbi, G. ; Firouzabadi, N.,Smart nanogels as promising platform for delivery of drug, gene, and vaccine; therapeutic applications and active targeting mechanism. Eur Polym J 2024,113400.” (Page 20, line 751-753)

Round 2

Reviewer 2 Report

Comments and Suggestions for Authors

The manuscript has been revised and improved. Therefore, it can be accepted for publication.